# Evaluation of anxiety level and the factors Affecting Anxiety in health care workers in Shahid Dr. Gholipour Hospital, Bukan, Iran during COVID-19 pandemic

Esmaiel Maghsoodi[1], Edris Hasanpour[2]¤, Farzaneh Soleimani[1], Moosa Aghal[1], Keyvan Mollarahimi[2]*

1 Department of Nursing, Maragheh University of Medical Sciences, Maragheh, Iran, 2 Department of Nursing. Boukan faculty of Nursing, Urmia University of Medical Sciences, Urmia, Iran

☯ These authors contributed equally to this work.
¤ Current address: Department of Nursing, Maragheh University of Medical Sciences, Maragheh, Iran
* keyvanmollarahimi@gmail.com

**Data Availability Statement:** relevant data are within the manuscript and its Supporting information files.

## Abstract

### Introduction

The rapid and dangerous spread of Covid-19 has caused psychological problems, especially anxiety among health care workers. Due to the lack of accurate information on the prevalence of anxiety and the factors affecting it among health care workers, especially in developing countries, this study aimed to investigate the level of anxiety and the effective factors in health care workers in Shahid Dr. Gholipour Hospital in Bukan, Iran in Covid-19 pandemic.

### Methodology

A descriptive cross-sectional study was carried out through census sampling among health care workers of the hospital. Totally, 358 participants entered the study. Data gathering tool consisted of a demographical information tool and Spielberger's Anxiety scale. To determine the effective variables on hidden and obvious anxiety use liner regression. significant predictors variable of anxiety was determined by logistic regression.

### Findings

Means score for obvious anxiety was 47.005 (5.4) CI: 46.47–47.54 and for hidden anxiety was 42.790 (4.88) CI: 42.43–43.29. Factors affecting participants' obvious anxiety based on the Unadjusted model were Work experience, Age group, Marital status and History of anxiety disorders and gender. In the Adjusted model, the age group and the anxiety history are considered as effective variables. One of the effective factors on hidden anxiety is the effect of gender variable in both Adjusted and Unadjusted models. Based on logistic regression test, job variables, age groups, marital status (are predicted variables of obvious anxiety. In hidden anxiety, marital status was introduced as a predictor variable.

**Funding:** This study was funded by Urmia University of Medical Sciences. The funders had no role in designing the study, collecting and analyzing the data, and preparing the manuscript. The main role of funders has been to issue licenses for access to samples in the field of study and to issue a code of ethics in the ethics committee of the institute.

**Competing interests:** The authors have declared that no competing interests exist.

## Conclusion

The majority of personnel experienced a high level of anxiety during the pandemic. Psychological examination and interventions are essential for the health care workers.

## Introduction

According to the definition of the ICD-11(International Classification of Diseases) "Generalized anxiety disorder is characterized by marked symptoms of anxiety that persist for at least several months, for more days than not, manifested by either general apprehension (i.e. 'free-floating anxiety') or excessive worry focused on multiple everyday events, most often concerning family, health, finances, and school or work, together with additional symptoms such as muscular tension or motor restlessness, sympathetic autonomic over-activity, subjective experience of nervousness, difficulty maintaining concentration, irritability, or sleep disturbance. The symptoms result in significant distress or significant impairment in personal, family, social, educational, occupational, or other important areas of functioning. The symptoms are not a manifestation of another health condition and are not due to the effects of a substance or medication on the central nervous system" [1]. Coronavirus appeared in Wuhan on the December 12th 2019 and rapidly spread to other cities of China and then the world. On the March 11th 2020, the WHO declared Covid-19 pandemic. By creating acute respiratory syndrome, the virus is highly contagious and spreads fast so that 4,010,834 are dead so far due to the virus in the world. Total number of deaths and diagnosed cases in the east Mediterranean region are 11,342,446 and 221,164 respectively. In Iran, 3,327,526 cases and 85,397 deaths have been recorded, which places Iran in 13th position in the world in terms of the spread of Covid-19 [2].

Covid-19 virus can be found in pulmonary secretion lavage, blood, and feces of the patients, which indicate the infection transmission paths [3]. According to studies, the main transmission path of the disease is through respiratory droplets (from the nose and mouth) and direct contact with contaminated surfaces. Incubation period of the disease ranges from 1–14 days, and mostly between 3–7 days [4]. Mean age of the patients ranges from 47–59 years and the prevalence in women is 41.9%–45.7% [5]. The symptoms are mostly fever, cough, and tiredness, and the course of disease in most of the patients is like that of influenza. A few of the patients experience respiratory distress and respiratory agony (through non-treatable metabolic acidosis, septic shock, and coagulation disorder) that lead to death [6]. All these highlight the risk to care-givers of Covid-19 patients [7].

The pandemic nature of the disease and its high mortality and transferability are the factors that make the disease a great concern for the public [8]. Studies show the irreversible physical and psychological effects of Covid-19 on health care providers. Anxiety is one of the most common psychological problems due to the high workload, seeing young critically ill patients and the unknown disease, which is probably prevalent among health workers as well as the general public [9]. Studies have been carried out about the effects of this disease on mental health of health workers. Anti-pandemic situation and heavy workload caused by this situation is a great risk to mental health of health workers. Therefore, it is essential to conduct psychological assessment and interventions on the patients and care-givers [4].

As suggested by studies, prevalence of SARS-COV-2 virus in health personnel and patients is 33–42% and 62–79% respectively. In fact, hospital environment is a main source of spreading SARS-COV-2 virus [10]. Pappa et al. In their review study, reported a prevalence of anxiety

among health workers 23.1 [11]. Taking into account the considerable risk caused by the disease for the public and medical team, the disease is a major cause of anxiety for hospital personnel. The medical teams are permanently exposed to the risk of disease given their long work hours and the tiredness caused by it [12]. Fighting at the front line of battle against Covid-19, the medical team sustains considerable physical and spiritual pains and as suggested by studies, they experience more severe mental stresses. In addition to the fact that there are no accurate statistics on the prevalence of anxiety among medical staff in Iran, unfortunately, the factors affecting its occurrence are also unclear [13]. Thereby, given the importance of anxiety and its notable effect on the life and work of medical personnel, the present study is an attempt to Evaluation of anxiety level and the factors Affecting Anxiety in Shahid Dr. Gholipour Hospital, Bukan-Iran (Is the capital of Bukan County, West Azerbaijan Province, Iran) during Covid-19 pandemic.

## Methodology

It should be noted for the satisfaction of the participants in the study due to the online of the questionnaire, first, how to do the work and the objectives of the study were explained in a separate page and making sure that people's information is confidential and is used for research purposes only. Individuals could enter the study and complete the questionnaire with their consent and if they were not satisfied, they could not complete the questionnaire and were excluded from the study. There were no legal minors in the study. The study was approved by the Ethics Committee of Urmia University of Medical Science (IR.UMSU.REC.1399.034) and also registered at Iran National Committee of Ethics in Biomedicine Researchers (https:// ethics.research.ac.ir).

The study was carried out as a descriptive cross-sectional work to measure anxiety and the factors in health care workers of Shahid Dr. Gholipour, Bukan, Iran (physicians, nurses, and Para clinical and administrative staff) during Covid-19 pandemic. The study population consisted of medical and administrative team of the hospital. The participants were selected through census sampling among 63 physicians, 382 nurses (nurse, health assistant, operation room nurse, and anesthetist), Para clinical staff in lab (n = 34) and radiology ward (n = 17) (totally 528 members). Eventually, 358 participants entered the study. The reason for choosing the census method for the subject is due to the existence of a hospital in the city and the involvement of all hospital staff and wards in Covid-19 disease. Due to the fear caused by the disease in the medical team and the availability of all of them, it was decided that all of them should be examined. The reason for the non-participation of some employees was due to sending an online questionnaire, which did not have the necessary enthusiasm to complete the questionnaire.

Inclusion criteria included desire to participate, an informed consent, at least high school diploma, working in the hospital, and Proper physical condition. Exclusion criteria were reluctance to participate in the study, incomplete questionnaire and having physical problems and other mental disorders.

Data gathering tools consisted of a demographics form including contact with Covid-19 patients, job (physician, nurse, Para clinical and administrative), work experience, age, gender, marital status, education, history of anxiety disorders, and history of using sedatives. "Spielberger questionnaire", which is a widely used tool to measure anxiety. The tool was designed in 1983 to examine anxiety based on hidden and obvious anxiety. The questionnaire has been translated into 30 languages and normalized for Iranian culture. It is comprised of 40 statements including 20 statements on obvious anxiety and 20 questions on hidden anxiety. Validity and reliability of the questionnaire were examined by Mahran and Cronbach's alpha was obtained equal to 0.81 with confidence level of 95% [14].

The obvious anxiety items are designed based on Likert's four-point scale (very low = 1, low = 2, high = 3, very high = 4). The highest score (4) indicates highest level of anxiety. Ten items out of the 20 items on obvious anxiety are scored inversely, which are items no. 1, 2, 5, 7, 10, 11, 15, 16, 19, and 20.

To obtain total score of anxiety for each respondent, given that some items are score inversely, total score of the 20 items is calculated. Therefore, hidden and obvious anxiety score ranges from 20 to 80. Anxiety in each part of anxiety: level from 31–20, Indicates mild anxiety, anxiety level from 32–42 Indicates moderate to low anxiety, rate 43–53 Indicates moderate to high anxiety, anxiety from 65–75 Indicates anxiety sever Anxiety and Anxiety level from 76 and above indicates anxiety It is very intense [15].

It should be noted for the satisfaction of the participants in the study due to the online of the questionnaire, first, how to do the work and the objectives of the study were explained in a separate page and making sure that people's information is confidential and is used for research purposes only. Individuals could enter the study and complete the questionnaire with their consent and if they were not satisfied, they could not complete the questionnaire and were excluded from the study. There were no legal minors in the study. Data analyses were one using Stata v.13 (Stata Corp, College Station, TX, USA) was significant level used (p ≤ 0.05). For descriptive variables frequency (%) and quantitative variable Mean (SD) reported.

According to the evaluation performed for regression defaults, the results showed that the defaults are confirmed and the conditions for regression are provided. These defaults are as follows:

1. Durbin Watson = 2.175 (Establishment of residual independence)

2. Absence of outlier (number of outlier 3 data), 5% ≥ number of outlier

3. VIF = 1.576–2.431 (no collinearity between variables)

4. Also, the normality of the residual distribution and the homogeneity of the residual variance were confirmed by a plot (Normal Probability Plot).

To determine the effective variables on hidden and obvious anxiety use liner regression (According to the default conditions (. significant predictors variable of anxiety was determined by logistic regression. The internal consistency of "Spielberger questionnaire" was suitable, with Cronbach's alpha being 0.73 among the 358 participants. According to the anxiety score of individuals and determining the cut off 40 for anxiety, people with a score above 40 were classified as a person with anxiety and people with a score below 40 as a person without anxiety and the dependent variable of anxiety (obvious and hidden) entered the regression model.

## Results

The results showed that out of 358 participants, 248 (69%) were from Covid wards (ICU and infectious wards), 96 (27%) were from other wards and 144 (40%) were from the administrative part. In addition, the results showed that 267 (74%) of the participants in the study were nurses, 52 (15%) were Para clinical staff, 26 (7%) were administrative staff, and 16 (4%) were physicians (Fig 1).

### Report by percentage

(Table 1) lists the demographics of the participants. Clearly, the majority of the participants are in 20–39 years old range. In terms of education, 84% of the participants had a bachelors'

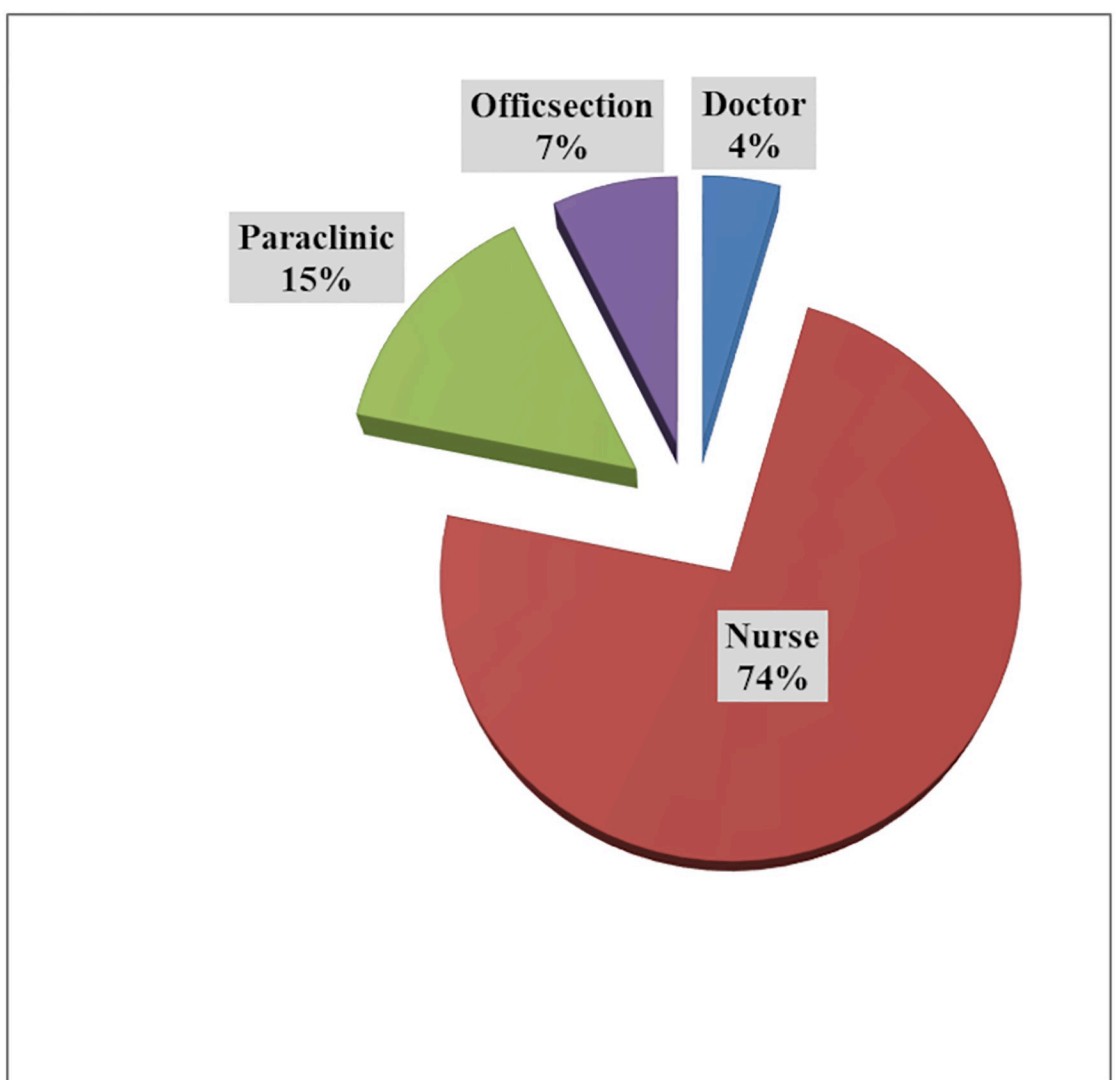

**Fig 1. The job status of participants in the study.**

degree and 58% had less than 10 years of experience. Participants are almost equal in terms of gender and 74 percent are married. Also, 16 percent of participants have a history of anxiety.

(Table 2) lists the scores of hidden and obvious anxieties. Severity of anxiety of the participants is listed in (Table 3). Mean and standard deviation of the score of obvious anxiety is 47.005 (5.4) CI: 46.47–47.54 and Hidden anxiety is 42.790 (4.88) CI: 42.28–43.29. Also, based on the severity of anxiety, 103 people have moderate to severe obvious anxiety and 184 people have moderate to severe hidden anxiety.

The evaluation of anxiety scores of the participants in the study by type of job showed that the highest level of obvious anxiety with an average of 47,538 (3,569) is related to administrative staff, followed by nurses with an average of 47,204 (5,375). In relation to hidden anxiety, the highest level of hidden anxiety with an average of 43.5 (4.349) is related to the staff of the Para clinic department, followed by nurses with an average of 42.776 (5.126). Based on the results, it was found that doctors have the lowest level of obvious and hidden anxiety (Table 4).

**Table 1. Demographic characteristics of the participants in the study of anxiety evaluation of the staff of Shahid Dr. Gholipour Hospital bukan in 2020.**

| Variable | Frequency (%) |
|---|---|
| **Gender** | |
| Male | 178 (49%) |
| Female | 180 (51%) |
| **Age group (year)** | |
| < 20 | 2 (0.6%) |
| 20–29 | 134 (37.5%) |
| 30–39 | 146 (40.7%) |
| 40–49 | 64 (17.8%) |
| 50–59 | 12 (3.4%) |
| **Marital status** | |
| Married | 264 (74%) |
| Unmarried | 94 (26%) |
| **Education** | |
| High school diploma | 14 (3%) |
| Bachelors' degree | 302 (84%) |
| Masters' degree | 30 (8%) |
| PhD | 12 (3%) |
| **Work experience (year)** | |
| < 5 | 108 (30%) |
| 5–9 | 102 (28%) |
| 10–14 | 64 (17%) |
| 15–19 | 50 (13%) |
| 20–24 | 20 (5%) |
| ≥ 25 | 14 (3%) |
| **History of anxiety disorders** | |
| Positive | 54 (16%) |
| Negative | 304 (84) |
| **History of using sedatives** | |
| Positive | 26 (7%) |
| Negative | 332 (92%) |

**Table 2. Total anxiety score of the participants in the study of anxiety evaluation of the staff of Shahid Dr. Gholipour Hospital bukan in 2020.**

| Variable | N | Mean (SD) | Min | Max | CI 95% |
|---|---|---|---|---|---|
| Obvious anxiety | 358 | 47.005 (5.4) | 47.005 (5.4) | 76 | 46.47–47.54 |
| Hidden anxiety | | 42.790 (4.88) | 42.790 (4.88) | 80 | 42.28–43.29 |

Factors affecting participants' obvious anxiety based on the Linear regression (Unadjusted model) were Work experience OR (-0.9) (P-Value ≤ 0.001, CI: -1.247 to -0.528), Age group OR (-2.05) (P-Value ≤ 0.001, CI: -2.663 to -1.448), Marital status OR (-1.81) (P-Value ≤ 0.003, CI: -3.012 to -0.608) and History of anxiety disorders OR (-2.04) (P-Value ≤ 0.007 CI: -3.524

**Table 3. Total severity of anxiety in the participants in the study of anxiety evaluation of the staff of Shahid Dr. Gholipour Hospital bukan in 2020.**

| Variable | Trivial | Average and below | Average and above | Relatively severe | Severe | Very severe |
|---|---|---|---|---|---|---|
| Obvious anxiety | | 69 (19.2%) | 72(72.2%) | 29 (8.1%) | | 2(0.6%) |
| Hidden anxiety | 4 (1.1%) | 170 (47.4%) | 175 (49%) | 8 (2.2%) | | 1(0.3%) |

**Table 4. Total anxiety score of the participants (depending on the type of job) in the study of anxiety evaluation of the staff of Shahid Dr. Gholipour Hospital bukan in 2020.**

| Job | Anxiety | N | Min | Max | Mean (SD) |
|---|---|---|---|---|---|
| Doctor | Obvious | 16 | 34 | 54 | 45.125 (5.402) |
| | Hidden | | 36 | 48 | 41.5 (3.829) |
| Nurse | Obvious | 264 | 23 | 76 | 47.204 (5.375) |
| | Hidden | | 28 | 80 | 42.776 (5.126) |
| Para Clinic | Obvious | 52 | 35 | 55 | 46.307 (4.399) |
| | Hidden | | 35 | 53 | 43.5 (4.349) |
| Office section | Obvious | 26 | 40 | 52 | 47.538 (3.569) |
| | Hidden | | 36 | 51 | 42.307 (3.739) |

to -0.563). that have a protective and decreasing role that with increasing each of these variables, the level of anxiety will decrease by one unit. Also, the gender variable OR (1.1) (P-Value ≤ 0.042, CI: 0.042 to 2.170) has an increasing role in the level of anxiety, which will increase with one unit of age. In the Adjusted model, the age group variable OR (-2.24) (P-Value ≤ 0.001, CI: -3.225 to -1.265) and the anxiety history variable OR (-1.98) (P-Value ≤ 0.007, CI: -3.419 to -0.560) are considered as effective variables (Table 5). One of the effective factors on hidden anxiety is the effect of gender variable in both Adjusted OR (1.379) (P-value ≤ 0.008 CI: 0.365 to 2.377) and Unadjusted OR (1.259) (P-value ≤ 0.016 CI: 0.228 to 2.281) (models (Table 6).

An analysis to identify the predictors variable anxiety revealed that job variables Doctor OR (6.53) (P-Value ≤ 0.021 CI: 1.485 to 9.784) nurse OR (5.46) (P-Value ≤ 0.011 CI: 1.769 to 7.348), Para clinic OR (2.68) (P-Value ≤ 0.047 CI: 1.023 to 5.685). in age group (20–29) year OR (0.016) (P-Value ≤ 0.001 CI: 0.002 to 0.112), age group (30–39) year OR (0.036) (P-Value ≤ 0.001 CI: 0.006 to 0.229), age group (40–49) year OR (0.088) (P-Value ≤ 0.01 CI: 0.014 to 0.561), marital status OR (0.034) (P-value ≤ 0.020 CI: 0.111 to 0.832) were significantly associated with the development of anxiety. In hidden anxiety, marital status OR (1.6) (P-Value ≤ 0.027 CI: 1.055 to 2.432) was the only significant predictor among all the hypothesized factors as a significant variable (Table 7).

**Table 5. Factors affecting obvious anxiety the staff of Shahid Dr. Ghalipour Hospital, Bukan in 2020.**

| Variable | Unadjusted | | Adjusted | |
|---|---|---|---|---|
| | B | P-Value (CI)* | B | P-Value (CI)** |
| Ward | -0.08 | 0.872 (-1.05 to 0.89) | | |
| Job | 0.143 | 0.743 (0.683 to 0.969) | | |
| Work experience | -0.901 | 0.001 (-1.247 to -0.528) | | |
| Age group | -2.055 | 0.001 (-2.663 to -1.448) | -2.245 | 0.001 (-3.225 to -1.265) |
| Gender | 1.106 | 0.042 (0.042 to 2.170) | | |
| Marital status | -1.810 | 0.003 (-3.012 to -0.608) | | |
| Education | -0.437 | 0.428 (-1/518 to 645) | | |
| History of anxiety disorders | -2.043 | 0.007 (-3.524 to -0.563) | -1.989 | 0.007 (-3.419 to -0.560) |
| History of using sedatives | -0.077 | 0.942 (-2.140 to 1.986) | | |

*p-value were calculated using the Linear regression

**Table 6. Factors affecting hidden anxiety the staff of Shahid Dr. Ghalipour Hospital, Bukan in 2020.**

| Variable | Unadjusted | | Adjusted | |
|---|---|---|---|---|
| | B | P-Value (CI) | B | P-Value (CI) |
| Ward | -0.019 | 0.968 (-0.939 to 0.902) | | |
| Job | 0.216 | 0.589 (-568 to 0.999) | | |
| Work experience | -0.030 | 0.871 (-0.395 to 0.335) | | |
| Age group | -0.233 | 0.455 (-0.844 to 0.378) | | |
| Gender | 1.371 | 0.008 (0.365 to 2.377) | 1.259 | 0.016 (0.228 to 2.281) |
| Marital status | -0.385 | 0.512 (-1.539 to 0.769) | | |
| Education | -0.590 | 0259 (-1.615 to 0.439) | | |
| History of anxiety disorders | -1.075 | 0.136 (-2.491 to 0.340) | | |
| History of using sedatives | -.309 | 0.757 (-2.266 to 1.649) | | |

*p-value were calculated using the Linear regression

## Discussion

The Covid-19 disease pandemic has plunged the world into a crisis that the result is psychological effects on all people [16]. The psychological effects of pandemics have always been of interest to researchers, and in the meantime, health care workers are at risk for disease and psychological complications due to the nature of the job and working conditions [17].

The results of this study showed moderate to severe anxiety in health workers during the outbreak of Covid-19 disease in the southern region of West Azerbaijan province. Type of job,

**Table 7. Predictors variable for hidden ana obvious anxiety in the staff of Shahid Dr. Ghalipour Hospital, Bukan in 2020.**

| variable | N | OR | P-Value (CI) * |
|---|---|---|---|
| **Obvious anxiety** | | | |
| **Job** | | | |
| Doctor | 16 | 6.535 | 0.021 (1.485 to 9.784) |
| Nurse | 264 | 5.467 | 0.011 (1.769 to 7.348) |
| Para clinic | 52 | 2.681 | 0.047 (1.023 to 5.685) |
| Officsection | | Reference | |
| **Age group (year)** | | | |
| < 20 | 2 | 0.87 | 0.9 (0.001 to 0.022) |
| 20–29 | 134 | 0.016 | 0.001 (0.002 to 0.112) |
| 30–39 | 146 | 0.036 | 0.001 (0.006 to 0.229) |
| 40–49 | 64 | 0.088 | 0.01 (0.014 to 0.561) |
| 50–59 | | Reference | |
| **Marital status** | | | |
| Married | 264 | 0.034 | 0.020 (0.111 to 0.832) |
| Unmarried | | Reference | |
| **Hidden anxiety** | | | |
| **Gender** | | | |
| Male | 178 | 1.602 | 0.027 (1.055 to 2.432) |
| Female | | Reference | |

*p-value were calculated using the Binary logistic

age group, gender, marital status, previous history of anxiety were factors affecting the level of anxiety of participants. In a way, the variables of type of job (Doctor, Nurse and Para Clinic), age group, and marital status and in obvious anxiety and gender in hidden anxiety revealed significant in regression logistic.

The prevalence of anxiety in health workers has been reported in studies with varying amounts. For example, in a systematic review of Salari et al., 25.8% were reported, or Lai et al. reported a prevalence of anxiety in health workers of 44.6%, indicating a low to moderate prevalence of anxiety which is almost consistent with the results of our study [9, 18]. The lowest prevalence was reported by Liu et al. With 16% in China [18] and the highest rate was reported by Shahin et al. With 60.2% in Turkey [19]. While the results of our study are consistent with the results of Zhu et al., Who reported 55.26% of self-reported anxiety in health workers, in general, the results of their study showed poor mental health in health workers and influential factors Recognized the psychological health of employees including education, income satisfaction, perceived respect for medical staff, frequency of verbal abuse, psychological tolerance in emergencies [20]. A study by Huang et al reported that the incidence of anxiety disorders was about 35.1%, and showed that people under the age of 35 and those who spent more than 3 hours a day thinking about Covid-19 disease were more likely to develop anxiety disorders. Nearly a quarter of health workers complain of sleep problems, which is significant compared to other occupations [21].

Although in their study, Gupta et al. (2020) reported mild anxiety at 50.8%, they reported a significant association between female gender and higher levels of anxiety as we do [22]. Gao et al., (2020) unlike our study, did not report a significant relationship between gender and the incidence of anxiety disorders. In our study, nurses had higher levels of anxiety than other health care workers [6]. The study by Lai et al. And Shahin et al. showed that, like our results, the nursing staff experienced the highest rates of depression and anxiety compared to other groups, but the Gupta study did not report a significant difference in the anxiety of nurses and physicians [18, 19]. Increased levels of anxiety are inevitable during an epidemic of infectious diseases. A study by Ruey Chen in Taiwan reported similar results at the time of the outbreak of SARS, with nurses being more anxious than before the outbreak [23].

In the study by Hacimusalar et al. (2020) In Turkey, the age range of the subjects was between 20 years. Among the medical staff, 42% were doctors, 34% were nurses and the rest were medical staff [24], in terms of age group, the participants in the study were similar to the present study, but most of the participants were physicians, which is different from the results of our study, where most of the participants were nurses. Also, according to the results, most of the participants were nurses, which is different from the results of Gupta et al. (2020) [22] and Mosolov et al. (2020) [25] studies in Russia, where most of the participants were physicians. In terms of gender ratio of participants in the study, the results indicate an equal ratio of men and women in the study, the results are different from studies [22, 25] where most participants were male.

A study by Alenazi et al. (2020) In Saudi Arabia showed that male gender, singleness, and low work experience were directly related to anxiety [26], which these results are consistent with the results of our study that male gender, singleness and low work experience were significantly associated with anxiety levels. Also in this study, the type of job was significantly associated with anxiety, which is different from our study. In the study by Shahin et al. (2020) in Turkey, female gender, age group, work experience and previous history of psychiatric illness were identified as factors affecting Covid-19 [19]. These results are not consistent with the results of our study, so that in our study these factors have been identified as a protective factor. In the study by Zhang et al. (2020) in China, the work department and direct contact with the patient were significantly associated with anxiety, while the results of our study did not

show such a relationship. Also, in their study, the level of education reduced anxiety, which our study showed no correlation.

In this study, marital status reduced anxiety and this is consistent with the results of Zhang's study [27]. The results of the study Mosolov et al. (2020) [25] are similar to the results of our study, in that age has been suggested as a protective factor, however, in this study, the physician's occupation was significantly associated with anxiety and did not agree with the results of our study. Examining the results, we find that a history of previous depression has played a protective role, it differs from the results of the study by Yang et al. (2020) [28] that a previous history of anxiety is significantly associated with anxiety.

The results of logistic regression in Hacimusalar et al. (2020) [24] showed that female gender has a significant relationship that it is different from the results of the present study that male gender has a significant relationship. In the study of Gupta et al. (2020) [22], the variables of female gender, singleness and type of education were considered as significant variables in the regression model, it differs from the results of the present study that the type of occupation, age group, marriage and male gender were significant. Evaluation of predictor variables in the study of Mosolov et al. (2020) [22] showed that the type of job has a significant relationship with anxiety, which is consistent with the results of our study and the type of job in the logistic regression model was significantly related.

Of course, an important issue such as the prevalence of Covid-19 in Iran, due to the conditions of sanctions and shortage of drugs, equipment and vaccines, unfortunately, in addition to affecting the general public, has a negative impact on hospital staff. Unfortunately, non-standard and quotas equipment and lack of financial resources are some of the problems that exist in Iranian hospitals, which in themselves cause some kind of anxiety and stress in hospital staff [9]. The dangerous nature of the disease and the existence of different mutated types of the virus puts double stress on the staff of health centers every day. On the other hand, the low rate of vaccination in Iran compared to developed countries, the publication of news related to the high prevalence of Covid in the country and on the other hand the decrease in prevalence in developed countries due to high vaccination population can also be sources of anxiety and other psychological disorders Be among the health care worker [13, 29].

## Conclusion

The results of the present study indicate a high level of anxiety among staff. Unfortunately, the country is now in the fifth peak and delta-type strains. Continuation of these conditions will lead to serious and worrying challenges. In this regard, strengthening the infrastructure, establishing new support conditions, developing and presenting training programs in order to comply with the necessary Corona disease conditions and vaccination of all personnel seems necessary. To ensure the continued effective work of these individuals, their mental health status should be monitored and timely and continuous interventions provided to support them. Pandemic psychological interventions include risk management and resilience training, effective methods for managing the psychological effects of medical staff. And the observance of hygienic principles (use of masks, prevention of gatherings, quarantine, and vaccination) and finally the obligation of all people to observe hygienic protocols, the result of which has a direct effect on reducing work stress and psychological stress of personnel.

## Supporting information

**S1 File.**
(SAV)

## Acknowledgments

The authors wish to express their gratitude to all personal and medical team members of Shahid Dr. Gholipour Bukan Hospital who supported this study.

## Author Contributions

**Formal analysis:** Keyvan Mollarahimi.

**Funding acquisition:** Moosa Aghal.

**Investigation:** Esmaiel Maghsoodi, Edris Hasanpour.

**Methodology:** Keyvan Mollarahimi.

**Project administration:** Farzaneh Soleimani.

**Software:** Keyvan Mollarahimi.

**Writing – original draft:** Keyvan Mollarahimi.

**Writing – review & editing:** Esmaiel Maghsoodi, Edris Hasanpour, Farzaneh Soleimani.

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
