## [Decision Letter · Decision Letter 0]

8 Mar 2021

PONE-D-20-38720

Examining anxiety level in medical team members in Shahid Dr. Gholipour Hospital, Bukan, Iran during COVID 19 pandemic

PLOS ONE

Dear Dr. mollararahimi,

Thank you for submitting your manuscript to PLOS ONE. After careful consideration, we feel that it has merit but does not fully meet PLOS ONE’s publication criteria as it currently stands. Therefore, we invite you to submit a revised version of the manuscript that addresses the points raised during the review process.

Thank you for submission in PLOS ONE. Manuscript has now been assessed by relevant experts. Referees find the draft interesting piece of work but raised major concerns in presentation of results, study instrument validity and reliability and methodological aspects. I will suggest authors to please consider the manuscript for revisions.

We look forward to receiving your revised manuscript.

Kind regards,

Tauqeer Hussain Mallhi, Ph.D

Academic Editor

PLOS ONE

Journal Requirements:

2. Please add the reference for teh Spielberger questionnaire.

3. In your Methods section, please provide additional information about the participant recruitment method and the demographic details of your participants. Please ensure you have provided sufficient details to replicate the analyses such as:   

-    a statement as to whether your sample can be considered representative of a larger population,

-    a description of how participants were recruited.

"In addition, the authors thank the Department of

Research and Technology of Urmia University of Medical Sciences for approving the research plan and

financial support."

"NO

Reviewers' comments:

Reviewer's Responses to Questions

**Comments to the Author**

1. Is the manuscript technically sound, and do the data support the conclusions?

Reviewer #1: Partly

Reviewer #2: No

Reviewer #3: Partly

Reviewer #4: Yes

Reviewer #5: No

2. Has the statistical analysis been performed appropriately and rigorously? 

Reviewer #1: Yes

Reviewer #2: No

Reviewer #3: Yes

Reviewer #4: Yes

Reviewer #5: Yes

3. Have the authors made all data underlying the findings in their manuscript fully available?

Reviewer #1: Yes

Reviewer #2: Yes

Reviewer #3: No

Reviewer #4: No

Reviewer #5: No

4. Is the manuscript presented in an intelligible fashion and written in standard English?

Reviewer #1: Yes

Reviewer #2: No

Reviewer #3: No

Reviewer #4: Yes

Reviewer #5: No

5. Review Comments to the Author

Reviewer #1: The research highlights an area which has been extensively covered by recent published literature including systematic reviews which are higher on the hierarchy of evidence. examples of recent published literature in similar settings and across the globe include:

Salari N, Hosseinian-Far A, Jalali R, Vaisi-Raygani A, Rasoulpoor S, Mohammadi M, Rasoulpoor S, Khaledi-Paveh B. Prevalence of stress, anxiety, depression among the general population during the COVID-19 pandemic: a systematic review and meta-analysis. Globalization and health. 2020 DJahanshahi AA, Dinani MM, Madavani AN, Li J, Zhang SX. The distress of Iranian adults during the Covid-19 pandemic–More distressed than the Chinese and with different predictors. Brain, behavior, and immunity. 2020 Jan 1.ec;16(1):1-1.

Xiong J, Lipsitz O, Nasri F, Lui LM, Gill H, Phan L, Chen-Li D, Iacobucci M, Ho R, Majeed A, McIntyre RS. Impact of COVID-19 pandemic on mental health in the general population: A systematic review. Journal of affective disorders. 2020 Aug 8.

Yıldırım M, Arslan G, Özaslan A. Perceived risk and mental health problems among healthcare professionals during COVID-19 pandemic: exploring the mediating effects of resilience and coronavirus fear. International journal of mental health and addiction. 2020 Nov 16:1-1.

Although the manuscript is scientifically sound it unfortunately doesn't add anything new or significant to existing knowledge base.

Reviewer #2: PONE-D-20-38720:

Introduction,

Line 5 in the introduction, what is the “O”

Line 11 in the introduction, check 41-9-45.7%

Methodology,

The design of the study is descriptive cross sectional not descriptive analytical

Line 7, it should be “participate” not “participant”

Regarding Spielberger questionnaire, where is the reference of the used questionnaire

“Validity and reliability of the questionnaire were examined by Mahran and P-value was obtained equal to 0.81 with confidence level of 95%.”. P-value is not correct in this sentence. It should be Cronbach’s alpha

Results,

Titles of the tables and figures are short and non-explanatory

Table (1): in education, the sum is 360 not 358. It should be 358

Table (4): the factors related to hidden or obvious anxiety??

Figure (1) is not clear at all

The results should include logistic regression analysis to determine the significant predictors of anxiety.

Discussion,

Deficient with no enough comparable results

Reviewer #3: Dear Authors,

The title is good area to recommend health care workers to prevention of anxiety. But the paper lacks certain analytic methods and coherence. Please respond accordingly from attached comments

Reviewer #4: This manuscript sheds light on an under-scrutinised issue since the onset of the COVID-19 pandemic, namely anxiety amongst frontline healthcare workers. It is a well-developed and executed study. Not unlike numerous Chinese studies examining anxiety and depression throughout the pandemic (1-4), this study adds to the literature on the mental health of frontline healthcare workers.

From a general perspective, this manuscript was well-written and the outline was clear and easy to follow. However, In the first paragraph of the introduction, one sentence reads “Total number of deaths and diagnosed cases in the east Mediterranean region are 605026 and 14024 respectively”. Shouldn’t this sentence read “Total number of diagnosed cases and deaths in the east Mediterranean region are 605026 and 14024, respectively”? In the final sentence of the paragraph, the phrase “so far” is mistakenly spelled “o far”.

The paragraph describing the Spielberger questionnaire, used to assess anxiety, in the Methods section could have been crafted better and definitely could have used some references. Additionally, the authors should expand on the specific tests utilised for their statistical analyses in the Methods section.

Overall, this manuscript was well-written and included a comprehensive suite of variables. It manages to shed light on factors contributing to anxiety amongst medical staff throughout the COVID-19 pandemic. Furthermore, it highlights the need for tailored psychological interventions for frontline healthcare workers deployed to COVID-19 wards in hospitals.

References

1. Zhu J, Sun L, Zhang L, Wang H, Fan A, Yang B, et al. Prevalence and influencing factors of anxiety and depression symptoms in the first-line medical staff fighting against COVID-19 in Gansu. Frontiers in psychiatry. 2020;11.

2. Zhang Z, Zhang L, Wang Y. COVID‐19 indirect contact transmission through the oral mucosa must not be ignored. Journal of Oral Pathology & Medicine. 2020;49(5):450-1.

3. Liu Y, Chen H, Zhang N, Wang X, Fan Q, Zhang Y, et al. Anxiety and depression symptoms of medical staff under COVID-19 epidemic in China. Journal of Affective Disorders. 2021;278:144-8.

4. Liu C-Y, Yang Y-z, Zhang X-M, Xu X, Dou Q-L, Zhang W-W, et al. The prevalence and influencing factors in anxiety in medical workers fighting COVID-19 in China: a cross-sectional survey. Epidemiology & Infection. 2020;148.

Reviewer #5: Abstract introduction Research gap was not indicated.

Data collection method was not clear main body materials and Methods Data analyses were not specified Results Outcome result was not stated Discussions

The finding was not contrasted.

6. PLOS authors have the option to publish the peer review history of their article (what does this mean?). If published, this will include your full peer review and any attached files.

Reviewer #1: No

Reviewer #2: No

Reviewer #3: **Yes: **Adem Abdulkadir Abdi

Reviewer #4: **Yes: **Shady Abdelsalam

Reviewer #5: No

---

## [Author Response · Author response to Decision Letter 0]

11 Aug 2021

hello and Regards all reviewers 

Sorry for the delay in submitting the re-edit and Thanks for the helpful comments.

Corrections and And explanation item in the revised manuscript and reviewers file.

with regarads

keyvan mollarahimi

---

## [Decision Letter · Decision Letter 1]

22 Oct 2021

PONE-D-20-38720R1Evaluation of anxiety level and the factors Affecting Anxiety in health care workers in Shahid Dr. Gholipour Hospital, Bukan, Iran during COVID-19 pandemicPLOS ONE

Dear Dr. mollararahimi,

Thank you for submitting your manuscript to PLOS ONE. After careful consideration, we feel that it has merit but does not fully meet PLOS ONE’s publication criteria as it currently stands. Therefore, we invite you to submit a revised version of the manuscript that addresses the points raised during the review process.

We look forward to receiving your revised manuscript.

Kind regards,

Tauqeer Hussain Mallhi, Ph.D

Academic Editor

PLOS ONE

Additional Editor Comments (if provided):

Dear Authors, Thank you for submitting the revising version. The revised draft have been assessed by referees again and found some more concerns which are needed to be addressed. Please consider the comments of the reviewer and respond the queries related to statistical analysis and tool validity.

Reviewers' comments:

Reviewer's Responses to Questions

**Comments to the Author**

1. If the authors have adequately addressed your comments raised in a previous round of review and you feel that this manuscript is now acceptable for publication, you may indicate that here to bypass the “Comments to the Author” section, enter your conflict of interest statement in the “Confidential to Editor” section, and submit your "Accept" recommendation.

Reviewer #3: All comments have been addressed

Reviewer #4: All comments have been addressed

Reviewer #5: (No Response)

2. Is the manuscript technically sound, and do the data support the conclusions?

Reviewer #3: Yes

Reviewer #4: Yes

Reviewer #5: Yes

3. Has the statistical analysis been performed appropriately and rigorously? 

Reviewer #3: Yes

Reviewer #4: Yes

Reviewer #5: Yes

4. Have the authors made all data underlying the findings in their manuscript fully available?

Reviewer #3: Yes

Reviewer #4: Yes

Reviewer #5: Yes

5. Is the manuscript presented in an intelligible fashion and written in standard English?

Reviewer #3: Yes

Reviewer #4: Yes

Reviewer #5: No

6. Review Comments to the Author

Reviewer #3: Comments in the first review are incorporated adequately. Try to rewrite results and discussion. Other issues were addressed

Thank You

Reviewer #4: Thank you for submitting this revised manuscript. I am satisfied that my earlier comments have been appropriately addressed.

Reviewer #5: The authors should state the appropriate mode they used for their data analysis. If logistic regression analysis was used, how?

Method should be clearly stated.

Have you checked internal consistency?

How did you assess your outcome of interest to use logistic regression model?

What are assumptions of your appropriate model for data analysis?

7. PLOS authors have the option to publish the peer review history of their article (what does this mean?). If published, this will include your full peer review and any attached files.

Reviewer #3: No

Reviewer #4: **Yes: **Shady Abdelsalam

Reviewer #5: No

---

## [Author Response · Author response to Decision Letter 1]

6 Dec 2021

Dear Dr. Tauqeer Hussain Mallhi and Reviewer #5:

Hi

Thank you for your attention, all issues raised in the final edition were corrected and resubmitted.

Keyvan Mollarahimi

Corresponding Author

Kind regards

---

## [Decision Letter · Decision Letter 2]

22 Feb 2022

Evaluation of anxiety level and the factors Affecting Anxiety in health care workers in Shahid Dr. Gholipour Hospital, Bukan, Iran during COVID-19 pandemic

PONE-D-20-38720R2

Dear Dr. mollarahimi,

We’re pleased to inform you that your manuscript has been judged scientifically suitable for publication and will be formally accepted for publication once it meets all outstanding technical requirements.

Kind regards,

Tauqeer Hussain Mallhi, Ph.D

Academic Editor

PLOS ONE

Additional Editor Comments (optional):

Thank you for revising the draft.

Reviewers' comments:

Reviewer's Responses to Questions

**Comments to the Author**

1. If the authors have adequately addressed your comments raised in a previous round of review and you feel that this manuscript is now acceptable for publication, you may indicate that here to bypass the “Comments to the Author” section, enter your conflict of interest statement in the “Confidential to Editor” section, and submit your "Accept" recommendation.

Reviewer #4: All comments have been addressed

Reviewer #5: All comments have been addressed

2. Is the manuscript technically sound, and do the data support the conclusions?

Reviewer #4: Yes

Reviewer #5: Yes

3. Has the statistical analysis been performed appropriately and rigorously? 

Reviewer #4: Yes

Reviewer #5: Yes

4. Have the authors made all data underlying the findings in their manuscript fully available?

Reviewer #4: Yes

Reviewer #5: Yes

5. Is the manuscript presented in an intelligible fashion and written in standard English?

Reviewer #4: Yes

Reviewer #5: Yes

6. Review Comments to the Author

Reviewer #4: (No Response)

Reviewer #5: The authors have addressed MAJORITY of the raised issues. Nevertheless, the authors need to refine through out the manuscript.

7. PLOS authors have the option to publish the peer review history of their article (what does this mean?). If published, this will include your full peer review and any attached files.

Reviewer #4: **Yes: **Shady Abdelsalam

Reviewer #5: No

---

## [Editor Report · Acceptance letter]

16 Mar 2022

PONE-D-20-38720R2 

Evaluation of anxiety level and the factors Affecting Anxiety in health care workers in Shahid Dr. Gholipour Hospital, Bukan, Iran during COVID-19 pandemic                                                                                               

Dear Dr. Mollarahimi:

I'm pleased to inform you that your manuscript has been deemed suitable for publication in PLOS ONE. Congratulations! Your manuscript is now with our production department. 

Kind regards, 

on behalf of

Dr. Tauqeer Hussain Mallhi 

Academic Editor

PLOS ONE